# Biomimetic System Based on Reconstituted Macrophage Membranes for Analyzing and Selection of Higher-Affinity Ligands Specific to Mannose Receptor to Develop the Macrophage-Focused Medicines

**DOI:** 10.3390/biomedicines11102769

**Published:** 2023-10-12

**Authors:** Igor D. Zlotnikov, Elena V. Kudryashova

**Affiliations:** Faculty of Chemistry, Lomonosov Moscow State University, Leninskie Gory 1/3, 119991 Moscow, Russia; zlotnikovid@my.msu.ru

**Keywords:** macrophage-based biochip, CD206, bronchoalveolar lavage, diagnostics

## Abstract

Progress in macrophage research is crucial for numerous applications in medicine, including cancer and infectious diseases. However, the existing methods to manipulate living macrophages are labor-intense and inconvenient. Here, we show that macrophage membranes can be reconstituted after storage for months at 4 °C, with their CD206 receptor selectivity and specificity being similar to those in the living cells. Then, we have developed a mannose ligand, specific to CD206, linked with PEG as an IR spectroscopy marker to detect binding with the macrophage receptor. PEG was selected due to its unique adsorption band of the C–O–C group at IR spectra, which does not overlap with other biomolecules’ spectroscopic feature. Next, competitive binding assay versus the PEG-bound ligand has enabled the selection of other higher-affinity ligands specific to CD206. Furthermore, those higher-affinity ligands were used to differentiate activated macrophages in a patient’s bronchoalveolar (BAL) or nasopharyngeal (NPL) lavage. CD206− control cells (HEK293T) showed only non-specific binding. Therefore, biochips based on reconstituted macrophage membranes as well as PEG-trimannoside as an IR spectroscopic marker can be used to develop new methods facilitating macrophage research and macrophage-focused drug discovery.

## 1. Introduction

Infections caused by pathogenic bacteria are one of the main causes of death in developing countries and a serious health problem in developed countries. The widespread use of antibiotics leads to the emergence and development of strains that are resistant and multi-resistant to several types of antibiotics at the same time, as well as reservoirs of latent or “dormant” bacteria [1,2,3,4,5,6,7]. Existing methods of early diagnosis and therapy in some severe cases may be ineffective. This is especially true of secondary atypical pneumonia (caused by *Mycoplasma pneumoniae* and *Chlamydia pneumoniae*), complicated bronchitis, and bronchial asthma, where viral and bacterial infection is complicated by an immune response, etc. Targeted delivery to macrophages is a promising approach to improve the effectiveness of drug therapy for respiratory diseases, the driver of which are macrophages: pneumonia, tuberculosis, etc. [8,9,10,11,12,13,14,15,16,17,18,19,20,21,22,23,24]. The CD206 receptor is expressed only on activated macrophages [2,25]. CD206 is a transmembrane protein, a C-type lectin that recognizes mannose, fucose, and *N*-acetylglucosamine residues of oligosaccharides of bacteria and pathogens for the subsequent absorption of this pathogen by the macrophage [8,16,22,23,24,26,27]. The perspective idea is to implement biomimetics of a polymer with carbohydrate labels as models for a pathogenic microorganism to develop and test drug delivery systems by targeting alveolar macrophages. At the same time, macrophages are interesting objects from the point of view of the diagnosis of infectious diseases: by their polarization status and phenotype, one can judge the etiology of the disease and choose the optimal treatment strategy.

Macrophages are part of innate immunity, the first line of defense against pathogens, so their activation is one of the key factors in a number of infectious diseases. For example, macrophages are important for the elimination of pathogens: *Mycobacterium tuberculosis*, HIV, and parasites such as leishmania and trypanosome, causing “sleeping sickness”, etc. [3]. At the same time, the type of macrophage polarization (M1 or M2) plays a key role in the course and features of the disease. Pro-inflammatory macrophages M1 are important for protection against bacterial infections, but bacteria can secrete substances that repolarize macrophages into anti-inflammatory M2, which is a condition for the formation of dormant resistant infection [28,29]. Repolarization of macrophages in tumors allows changing the microenvironment from “cold” to “hot”, thereby activating the immune response against the tumor [30,31,32]. Excessive pro-inflammatory activity of macrophages can also be dangerous (for example, cytokine storm in COVID-19) [18]. Thus, for medical and bioanalytical applications, it is important to create methods for the express analysis of activated macrophages. On the other hand, determination of the effectiveness of binding of specific ligands to mannose receptors of macrophages is necessary for the development of effective delivery systems of targeted action to increase the effectiveness of therapy.

The existing methods to manipulate living macrophages are labor-intense and inconvenient. Here, we developed the bioanalytical system based on the macrophage-derived membranes, which can be reconstituted after drying on a cell film and storage for months at 4 °C, with their CD206 receptor selectivity and specificity similar to those in the living cells.

We considered 3 analytical applications of CD206 receptor-based biosensing. The first one is targeted delivery systems of the drugs to macrophages, which we considered in a series of our papers [22,23,24,33,34,35,36,37,38]. The development of the macrophage CD206 receptor-based biosensor would provide an effective testing system for optimizing of the structure of the specific ligands to alveolar macrophages for the therapy of the macrophage-associated diseases.

The second possible application is as follows: macrophage-based biosensors as a selective recognizer of carbohydrate fragments of pathogenic microorganisms may provide promising methods for the early diagnosis of diseases, determination of the intensity of inflammation, and efficiency of the treatment. Examples of experimental confirmations of the effectiveness of macrophage-based sensors are described in the literature: a sensitive and simple electrochemical sensor of mouse macrophage cells was developed for the early detection of lipopolysaccharides to assess the toxicity of pathogenic bacteria [39]. However, apparently, for such methods, significant signal distortion can be observed due to high impact of a non-specific signal.

In contrast, the CD206-targeting system suggested here works with receptors expressed mainly by disease-causing activated macrophages. A biosensor with deposited CD206+ macrophage membranes has a number of advantages: (1) high selectivity to oligomannosidic ligands (*K*_d_ 10^−7^–10^−9^ M), (2) expressiveness of the analysis, (3) the possibility of multiple use, and (4) stability during storage.

The third application is the development of biosensors based on specific polymer ligands to CD206 macrophage receptors to diagnose the status of polarization and activation of macrophages in biological fluids (for example, bronchoalveolar lavage—BAL [40,41,42,43,44]) to monitor the course of treatment and the ability to adjust the course of therapy. Polymers with oligo- and polymannoside fragments can be used to detect activated macrophages and determine their M1–M2 polarization status. These are prerequisites for the development of personalized medicine.

Thus, the main idea of the work is to develop biosensors for analyzing the effectiveness and specificity of ligand binding with polarized macrophages. To do this, we synthesized an IR marker that specifically binds to the mannose receptors of macrophages. By displacing such an IR marker, the affinity for CD206 of various polymer oligosaccharide ligands with different molecular architecture can be determined. We tested our specific ligands on living macrophage cells and compared data on binding specificity on macrophage membranes. Our second bioanalytic system consists of the determination of the polymer binding degree with bronchoalveolar lavage (BAL) (which contains activated macrophages) and nasopharyngeal lavage (NPL) (as negative control that does not contain activated macrophages).

This work opens up experimental foundations for the creation of highly effective biosensors for the early diagnosis of macrophage-associated diseases, for monitoring the course of therapy and personalized medicine.

## 2. Materials and Methods

### 2.1. Reagents

Mannan (46 kDa), polyethyleneimine 1.8 kDa (PEI1.8), 2-hydroxypropyl-β-cyclodextrin (HPCD), FITC, D-mannose, D-galactose, spermine, putrescine, NaBH_3_CN, and activated PEG 5 kDa (N-succinimidyl ester of mono-methoxy poly(ethylene glycol)) were purchased from Sigma Aldrich (St. Louis, MO, USA). Mannotriose-di-(N-acetyl-D-glucosamine) (triMan) was obtained from Dayang Chem (Hangzhou, China) Co., Ltd. Carbonyldiimidazole (CDI) was obtained from GL Biochem Ltd. (Shanghai, China). Other chemicals (salts and acids) were from Reakhim Production (Moscow, Russia).

### 2.2. Polymer Synthesis and Characterization

Activated HPCD was obtained as described earlier [33,45] by the reaction of HPCD OH-groups with carbonyldiimidazole in DMSO.

#### 2.2.1. Mannan-sp-HPCD

A 100 mg mannan sample was dissolved in 10 mL of 1 mM HCl; then, a 0.1 molar excess of KJO_4_ relative to mannose units was added. The mixture was incubated for 30 min at 40 °C. Periodate purification was performed using dialysis (cut-off 3 kDa) against water for 2 h. Ten milligrams of spermine and 5 mg of NaBH_3_CN were added to the mixture at controlled pH 4. Then, the mixture was incubated for 6 h at 50 °C followed by purification by dialysis against water (cut-off 3.5 kDa) for 6 h. Additionally, 35 mg (per HPCD) of activated HPCD was added to the mixture, and incubation was continued at the same temperature for 6 h. The final purification of samples was performed using 24 h dialysis (cut-off 6–8 kDa).

#### 2.2.2. HPCD-PEI-X

FITC labeling of PEI was performed at the first step for parts of the samples; the rest of the polymers were unlabeled.

To the aqueous solution of PEI (2 mL of 50 mg/mL in 0.01 M HCl), a solution of FITC (10 mg in 1 mL DMSO) was added drop by drop with stirring; the pH was brought to 8. The mixture was incubated at 50 °C for 2 h, followed by purification by dialysis against water (cut-off 1 kDa) for 12 h. Then, to the PEI–FITC solution (or free PEI), activated HPCD was added at 6-fold molar excess. The mixture was incubated at 40 °C for 12 h at pH 7.4 (PBS) followed by dialysis against water (6–8 kDa) for 6 h. The sample was divided into three equal parts, to which an aqueous solution of (1) mannose (15-fold times molar excess over PEI), (2) triMan (10-fold molar excess over PEI), and (3) galactose (15-fold molar excess over PEI) were added, respectively. NaBH_3_CN has been added to each of them. The samples were incubated for 24 h at pH 5 and 50 °C followed by purification using dialysis (cut-off 6–8 kDa) for 12 h.

The final purification of the samples was performed by HPLC gel filtration in a Knauer chromatography system (Knauer, Berlin, Germany) on a Diasfer-110-C18 column (BioChemMack, Moscow, Russia) [23]. The concentration of the polymers was checked using CD spectroscopy (Jasco J-815 CD Spectrometer, JASCO Corp., Tokyo, Japan).

All samples were freeze-dried for two days at −60 °C (Edwards 5, BOC Edwards, Burgess Hill, UK). The degree of mannosylation was calculated according to spectrophotometric titration of amino groups (before and after mannosylation) with 2,4,6-trinitrobenzenesulfonic acid.

Determination of the hydrodynamic diameter of the synthesized polymers was carried out by NTA (Nanoparticle Tracking Analysis) using a Nanosight LM10-HS device (Nanosight, Amesbury, UK) [46].

### 2.3. IR Marker Synthesis and Characterization

Samples of 20 mg of triMan, 7 mg of putrescine, and 5 mg of NaBH_3_CN are mixed and dissolved in 1 mL of 1 mM HCl. Then, the mixture was incubated at 50 °C for 6 h, followed by purification by dialysis against water (cut-off 1 kDa). Seventy milligrams of activated PEG in PBS (pH 7.4) was added to the triMan–putrescine product; the mixture was incubated for 12 h at 50 °C, followed by purification by dialysis against water (cut-off 3.5 kDa). The sample is dried and characterized as indicated in the section above: radius 90 ± 10 nm, ζ-potential −2.0 ± 0.3 mV, and molecular weight 6.1 ± 0.3 kDa.

### 2.4. Biosensors Based on CD206+ Macrophages to Determine the Affinity of Polymer Carbohydrate Ligands

#### 2.4.1. CD206+ Macrophage and CD206− HEK293T Cultivation

Human monocyte cell line THP-1 was used. Macrophage-like cells were derived from THP-1 cell line as was described before [47]. Cells were obtained from the bank of cell lines of Lomonosov Moscow State University. THP-1 cells were cultured on RPMI-1640 (Gibco, Carlsbad, CA, USA), supplemented with GlutaMAX™ Supplement (Gibco, Carlsbad, CA, USA), and buffered with 10 mM HEPES pH 7.4 containing 10% heat-inactivated FBS (Gibco, Carlsbad, CA, USA) and 1% antimycotic antibiotic (HyClone Laboratories, Inc., Logan, UT, USA) at 37 °C and 5% CO_2_. Macrophage-like cells were derived by adding to THP-1 (0.5 million cells/mL) 100 nM phorbol 12-myristate 13-acetate (PMA, p8139, Sigma Aldrich, St. Louis, MO, USA) for 72 h. After 72 h, the medium was replaced, and the cells were cultured for another 96 h.

As a control CD206-negative system, we used the embryonic kidney human epithelium cells (HEK293T) that were cultured in a DMEM with 4.5 g D-glucose (Life Technologies, Carlsbad, USA) supplemented with 10% fetal bovine serum (FBS) (Gibco, USA) and 100 units/mL of penicillin and streptomycin. Cell passaging reached a 70–90% confluent monolayer. The following conditions are maintained in the incubator: temperature 37 °C, 5% CO_2_ in air at constant humidity. Removal of cells from culture plastic is carried out using 0.05% trypsin/EDTA solution (Hyclone, Logan, UT, USA).

#### 2.4.2. Cytometry and Immunocytochemistry

Dry samples of polymers conjugated with an FITC fluorescent label were diluted in PBS (PanEco, Moscow, Russia) with 10% DMSO (Sigma-Aldrich, St. Louis, MO, USA). Macrophage-like cells were washed with Hanks’ solution (PanEco), and polymers were added in serum-free media. For cytometry assay, cells were incubated for 40 min, washed thrice with Hank’s solution, and scraped in Versen’s solution. Cell suspension was centrifuged at 300× *g* for 5 min at +4 °C. The cell pellet was resuspended in a PBS solution. For immunocytochemistry assay, cells were incubated for 4 min, washed with PBS, and fixed with 4% formaldehyde solutions (Panreac, Darmstadt, Germany).

Cytometry was performed on a BD FACSAria™ III Cell Sorter instrument. The fluorescence of the FITC-conjugated polymer captured by the cells was analyzed. Immunocytochemistry was performed as earlier described [23,34].

#### 2.4.3. Confirmation of Macrophage Differentiation and CD206 Expression and HEK293T Cell Characteristics as a Negative Control

Macrophages were immunocytochemically characterized using confocal microscopy as well as by cytometry assay. Non-specific binding sites were blocked using 10% solution of normal goat serum in PBS with 1% bovine serum albumin for 1 h at 22 °C. CD206 labeling was performed using anti-CD206 antibodies (ab64693, Abcam, Cambridge, MA, USA, 1:100) or rabbit polyclonal control IgG (910801, Biolegend, San Diego, CA, USA) as a control for 2 h at 22 °C and subsequently with goat–anti-rabbit antibody conjugated with Alexa594 (A11037, Molecular Probes, Invitrogen, Waltham, MA, USA, 1:1000). The nuclei were stained with DAPI. Samples were analyzed with a Leica DM6000B fluorescent microscope equipped with a Leica DFC 360FX camera (Leica Microsystems GmbH, Wetzlar, Germany). Most of these cells were CD206-positive according to immunocytochemical analysis. CD206-positive cells effectively phagocytosed predominantly high-affinity polymeric conjugates with trimannoside or mannan: cytometry assay determined that ~80% macrophage-like cells were FITC-positive after adding specific HPCD-PEI1.8-triMan ligands, and ~55% were FITC-positive after adding monomannose ligand HPCD-PEI1.8-Man.

Embryonic kidney human epithelium (HEK293T) cells were teased as well. Cells were characterized by cytometry assay. HEK293T cells showed non-significant mannoside polymer uptake. Cytometry assay determined that ~15–20% HEK293T cells were FITC-positive after adding both HPCD-PEI1.8-triMan and HPCD-PEI1.8-Man ligands.

#### 2.4.4. CD206+ Macrophage Membrane-Based Sensors

Macrophage cells were placed on a 24-well culture plate after activation and verification of CD206 expression by flow cytometry as described above. The cells were washed with PBS buffer pH 7.4; then, part of the cells was incubated with polymeric ligands for 15–180 min, washed with PBS buffer pH 7.4 (three times), and analyzed for ligand binding capacity.

Other part of the cells was left untreated. The cells were washed with PBS buffer pH 7.4. Then, the liquid was removed, and samples were treated with gentle flow of dry air and used immediately or stored for 1–3–6 months at 4 °C.

The binding capacity of the CD206+ macrophage membrane was determined using FTIR and fluorescence spectroscopy. The CD206+ macrophage membrane (in a 24-well plate) was rehydrated by 30 min incubation in PBS solution. Then, incubation with polymer samples (mannosylated polymers, FITC-label or IR-marker) for 5–60 min followed by analysis of both the solution and macrophage membrane suspension (after cells detaching from the plate with Versin solution) was carried out. The binding capacity after 1–3 months of storage decreased by no more than 30%.

### 2.5. Bronchoalveolar and Nasopharyngeal Lavage Studies

Bronchoalveolar and nasopharyngeal lavages were provided by one of the authors of the work with inflammation of the respiratory tract (bronchitis) of medium degree. The isolation was carried out according to the method described in the work [48]. Five milligrams of *N*-acetylcysteine was added to them to dilute mucus, followed by washing with PBS three times (1000 g, 5 min). The following were the characteristics of the BAL studied: the content of neutrophils 3.4%, alveolar macrophages 87.4%, lymphocytes 9.0%, and eosinophils 0.2%. BAL suspension containing macrophages or NPL (as negative control) was incubated with polymer samples for 5–60 min. FTIR spectra of lavage + polymer or fluorescence spectra of FITC-labeled polymers were recorded during or after the process.

### 2.6. FTIR Spectroscopy

FTIR spectra of samples were recorded using a Bruker Tensor27 spectrometer equipped with a liquid nitrogen-cooled MCT (mercury cadmium telluride) detector, as described earlier [33,35,46,49].

### 2.7. UV-vis Spectroscopy and CD Spectroscopy for the Characteristics of the Ligand Spectral Properties and Modification Degree

UV-vis spectra of FITC-labeled polymers were recorded on an UltraSpec 2100 pro device (AmerSham Biosciences, Cambridge, UK) to quantify the FITC amount. The CD spectra of carbohydrate-labeled polymeric ligands were recorded on a Jasco J-815 CD Spectrometer (JASCO Corp., Tokyo, Japan).

### 2.8. Fluorescence Spectroscopy

The fluorescence emission spectra of FITC-labeled non-adsorbed polymer solutions were recorded on a Varian Cary Eclipse fluorescence spectrometer (Agilent Technologies, Palo Alto, CA, USA). λ_exci_ = 490 nm and λ_emi_ = 515 nm.

## 3. Results and Discussion

### 3.1. General Design of Investigation

This work is devoted to the creation of experimental bases of biosensors for diagnosing the status of the disease of patients. To achieve this goal, we consider CD206+ macrophages as a target for the delivery of antibacterial (or anticancer, etc.) drugs and as cells of the immune system playing a key role in inflammation, that is, by which the disease can be characterized. With the use of polymer ligands with carbohydrate residues, it is possible to monitor the status of the disease: determine the degree of inflammation, determine the effectiveness of the treatment, and select the appropriate medicine (personal therapy). Thus, our study consists of two blocks: (1) development of a prototype of the biochip, which is a system with deposited macrophages, and (2) study of the effectiveness of the mannosylated polymeric system interaction with bronchoalveolar and nasopharyngeal lavages.

The first block includes the development of a biochip prototype—a system that consists of macrophage membranes deposited on a well plate and the study of their binding capacity with mannosylated polymers using fluorescence spectroscopy as well as FTIR spectroscopy. To determine the selectivity of the recognition of polymers with carbohydrate labels, we studied macrophage membranes in comparison with the macrophages themselves, and also non-phagocytic HEK293T cells as negative control.

The second block includes the study of the interaction of polymeric ligands with different affinity to macrophages derived from bronchoalveolar or nasopharyngeal as a way of indicating macrophage-associated disease and the effectiveness or ineffectiveness of the applied strategy.

### 3.2. Biosensors Based on CD206+ Macrophages

#### 3.2.1. Synthesis and Characterization of Polymers Recognized by Biosensors

To implement the biosensor concept, we synthesized an IR marker based on a combination of two components: triMan (a high-affinity ligand for macrophages), covalently attached to PEG, which gives a high-intensity conservative band in the IR spectrum that practically does not overlap with others (Figure 1). When testing the affinity and specificity of ligands, the IR marker will compete for binding with ligands of different molecular architectures. Therefore, we synthesized 4 mannosylated polymers (mimicking the carbohydrate patterns of bacteria) with different affinity to CD206+ macrophages (Table 1). This original technique was first presented by us as an alternative for fluorescent, radio labels, and others.

The synthesis of the IR marker is carried out in 2 stages: reductive amination of the Schiff base formed from the amino group of putrescine (spacer) and the aldehyde group of the reducing end of the trimannoside derivative (triMan) followed by PEGylation. The FTIR spectrum of the triMan–PEG conjugate (Figure 1) shows separated peaks corresponding to C–O–C oscillations of both components (1025 cm^−1^ for triMan and 1080 cm^−1^ for PEG), intense bands of C–H bond oscillations in PEG (2885 and 2920 cm^−1^), as well as deformation oscillations of the N–H in amide bond (blue line: 1560 cm^−1^) instead of a peak at 1705 cm^−1^ (red line) corresponding to the activated PEG ether, which will confirm the success of crosslinking.

In addition to the IR marker, we synthesized and tested 4 polymers (Table 1) with different affinity to macrophages (as we have previously shown in a series of studies on a model receptor ConA and directly on living CD206+ macrophages): a polymer containing a label based on triMan or mannan exhibits high affinity to macrophages (Table 1); a carrier with a monomannose label (Man) that binds with medium efficiency; and a galactose labelled carrier (GAL) that show weak affinity. We used polymeric ligands: mannan, modified cyclodextrin (HPCD), and polyethylenimine grafted with HPCD modified by triMan, Man, and Gal carbohydrate labels.

For biochip prototype development, we use CD206+ macrophage-derived membranes (membranes of macrophages dried on a polystyrene plate—then rehydrated and incubated with ligands). Macrophages are a difficult-to-grow cell culture that can be studied for several hours. On the contrary, macrophage dried membranes can be considered as an analytically significant, stable, and robust model. Indeed, it turned out that CD206+ macrophage membranes demonstrate a binding ability and ligand specify similar to original cells (Table 1). The biochip prototype based on macrophage dried membranes shows rather good stability during storage. The ligand binding capacity decreased by ~27% in 1 month and by ~31% in the next two months.

#### 3.2.2. Biosensing of Polymers Mimicking with Bacteria by Membranes of CD206+ Macrophage Phagocytic Cells

FTIR spectroscopy provides valuable data on the interaction of cells with polymer systems, including the possibility to study the molecular mechanism of recognition. Here, we have developed an original technique for detecting the selectivity of the action of drug formulations using FTIR. With regard to biosensing by macrophage membranes, we expect that FTIR spectroscopy will become a tool for studying the affinity of the receptor–ligand interaction.

Figure 2 shows the FTIR spectra of CD206+ macrophage membranes that were preincubated with a medium affinity polymer with a monomannose label (HPCD-PEI-Man) when displaced by various excesses with a highly active triMan–PEG IR marker. Therefore, the criterion for evaluating the affinity and specificity of the tested ligands is the concentration of semi-displacement of this medium affinity polymer with monomannose label. When adding a PEG-triMan (IR marker) to the system macrophage with HPCD-PEI-Man, an increase in the intensity of the characteristic peaks of the IR marker is observed in the spectra (Figure 2b): 2924 cm^−1^ (ν_as_ CH_2_ groups), 2853 cm^−1^ (ν_s_ CH_2_ groups), 1600–1700 cm^−1^ (Amide I), 1500–1600 cm^−1^ (Amide II), and a specific band at 1080 cm^−1^ (ν C–O–C of PEG). The increase in intensity is due to the adsorption of the IR marker on the macrophage membrane: the IR marker gives a more intense absorption band in the IR spectrum compared with HPCD-PEI-triMan.

To control the specificity (non-adhesion binging of the polymer), the reverse process of displacement of the triMan–PEG IR marker with the addition of the mannosylated polymeric ligands was studied. Therefore, Figure 2c shows the FTIR spectra of CD206+ macrophage membranes with the pre-adsorbed triMan–PEG IR marker during displacement by the high-affinity ligand HPCD-PEI-triMan. Since the fraction of the adsorbed IR marker decreases, the opposite picture is observed: the decrease in the intensity of the absorption band of the CH_2_ groups (2924 and 2853 cm^−1^), to a certain level corresponding to equilibrium in 15 min. This means that the test system is working; the higher-affinity (multivalent trimannoside) ligand displaces the monovalent triMan–PEG marker.

Quantitatively, the recognition and binding efficiency of polymers by biochips with macrophages (dry CD206+ macrophage cells stored for 3 months) was studied using fluorescence spectroscopy using the FITC label (Table 2). The high-affinity polymer HPCD-PEI-triMan-FITC firmly binds to macrophage membranes on which a Gal- or Man-labeled polymer (low and medium affinity to CD206, respectively) was pre-adsorbed. However, at the same time, the binding of HPCD-PEI-triMan-FITC is inhibited by mannan- and triMan-containing polymers, which proves the binding ability of biochips with macrophages, with the involvement of CD206 receptors. Low-affinity HPCD-PEI-Gal-FITC binds only to macrophage membranes on which HPCD-PEI-Gal is adsorbed; other ligands inhibit binding. Thus, macrophage membranes retain CD206 mediated binding capacity (specific to oligomannoside skeletons) of ligands mimicking bacterial patterns.

#### 3.2.3. Negative Control for Macrophage Biosensor—Binding of Mannosylated Polymers by HEK293T Non-Phagocytic CD206− Cells

As a negative control for macrophages, we selected non-phagocytic HEK293T (CD206−) cells to show the specificity of the developed biochips. Figure 3 shows the FTIR spectra of HEK293T cells incubated with polymers (0–30 min). We observe an increase in the intensity of characteristic peaks due to non-specific adsorption of the polymer, in particular, amide I, responsible for protein–ligand binding. In the case of HEK293T cells, a different situation with selectivity of ligand binding is observed compared with CD206+ macrophage cells. The triMan ligand does not show greater affinity than Man- and Gal-. On the contrary, the ligands show binding to the HEK293T cells in the following order: HPCD-PEI-Man—70%, HPCD-PEI-Gal by 56%, HPCD-PEI-triMan—55%, Mannan-sp-HPCD—55%. Selectivity index ~1. While in the case of CD206+ cells, Mannan-sp-HPCD and triMan were the most affine ligands, for them, the dramatic difference (more than 200% compared with non-specific Gal-ligand (Figure 2); so the selectivity index is ~2) is observed. Thus, FTIR spectroscopy confirms the non-selectivity and non-specificity of the adsorption of polymers with carbohydrate labels on the HEK293T surface. Quantitative data of polymer adsorption on HEK293T (CD206−) cells were obtained also using fluorescence spectroscopy (Figure 4). Weak adsorption is observed for HEK293T (4–10% depending on the concentration of the polymers), while almost the same was observed for all (mono- and multimannose) labels studied (non-selective ligand binding). Therefore, only CD206+, i.e., macrophages, selectively bind target oligomannoside-containing polymers.

### 3.3. Binding of Polymers with Bronchoalveolar and Nasopharyngeal Lavage

The second biosensor system developed here is designed to analyze BAL by determining the ratio (Index) of the degree of binding of specific (triMan) and non-specific ligands (Man and Gal) with macrophages—as a criterion of the degree of inflammation (or treatment efficiency). The test system is arranged as follows: a suspension of macrophages isolated from BAL is added to the polymer ligand, and after incubation, the degree of binding of the high- and low-affinity ligand with BAL macrophages is determined to find the CD206-mediated specific index. Bronchoalveolar lavage (BAL) contains a large number of macrophages (85–90% of the cells), which have overexpressed CD206 receptors in the case of inflammation. The following were the characteristics of the BAL studied: the content of neutrophils 3.4%, alveolar macrophages 87.4%, lymphocytes 9.0%, and eosinophils 0.2%. We considered this substance as analytically significant for monitoring the course of treatment and the selection of personalized medicine. Nasopharyngeal lavage (NPL) contains excess of mucin, which binds with all of oligo- and polysaccharide polymers. It is chosen as a control biofluid for BAL since it practically does not contain activated macrophages. Table 3 presents comparative characteristics of the absorption of FITC-labeled polymers with BAL and NPL. Apparently, BAL selectively binds the polymer with triMan- or mannan label. On the contrary, for monomannose ligands, binding is inhibited by mannan (selectivity index is approximately 2), which indicates the polymer–macrophage interaction at the CD206 active site. The values of the fraction of the absorbed polymer are small, since it is specifically bound only by activated macrophages (the content of which is not too high in BAL samples from a patient with moderate inflammation). In the case of NPL containing mucin, we observe a non-specific bonding of a large number of carbohydrate polymers. Binding is practically not inhibited by mannan; on the contrary, it increases, probably due to the additional capture of polymers by mannan adsorbed on mucin. Thus, BAL can be considered as an object for the determination of CD206+ macrophages in disease and correction of therapy.

Complementary data on the interaction of BAL with polymers are provided by FTIR spectroscopy. Figure 5 shows the FTIR spectra of a suspension containing alveolar macrophages during online incubation with polymers with different affinity to CD206 receptors. The main peak sensitive to the protein–ligand interaction is Amide I, changes of which correlate with the efficiency and binding mechanism. For the non-specific Gal-modified polymer, the changes are almost linear (Figure 5b,e), which indicates non-specific binding and involvement of other receptors. For a mono-Man-modified polymer, a similar pattern is observed, but with a large hyperbole character and an intensity of changes of 13% (Figure 5a,e)—mixed binding (specificity + non-specificity). TriMan- and mannan-containing polymers demonstrate highly selective interaction with macrophage mannose receptors, since significant changes in the intensity of Amide I (up to 50%) and the hyperbolic form of the saturation curve are observed (Figure 5c–e). Thus, BAL selectively binds polymers targeted at CD206+ macrophages, which is an experimental basis for the primary screening of the type of disease (macrophage-dependent or not, CD206+ or −).

With FTIR spectroscopy, a suitable analysis time (of 20–45 min) was determined. During this time, equilibrium is reached in the system, and those ligands that were supposed to be absorbed are absorbed. The output of the saturation curve indicates the correctness and stability of the target signal in the time interval of 20–45 min.

## 4. Conclusions

Biosensors based on macrophage membranes are of great interest for bioanalytical applications (screening of ligand affinity, access systems, study of macrophage polarization) and, most importantly, as a model of macrophages for experiments without culturing living cells. We have developed a prototype of a biochip containing dried membranes of CD206+ macrophages that maintain the binding capacity of a CD206+ mannose receptor. Using FTIR and fluorescent spectroscopy, we demonstrated the specificity of CD206+ recognition by macrophages of ligands carrying oligo- and polymannoside skeletons and, at the same time, showed the non-specificity and low binding capacity of non-phagocytic CD206 HEK293T cells. As the second analytical application of the test system, we offer screening of interactions of bronchoalveolar lavage (BAL) with specially developed polymers with carbohydrate labels to study macrophage dependence and the CD206-mediated character of bacterial or other disease. This will make it possible to clarify the status of diseases, to study the effectiveness of the drug on BAL, in order to apply personalized therapy to achieve the best treatment effect.

## Figures and Tables

**Figure 1 biomedicines-11-02769-f001:**
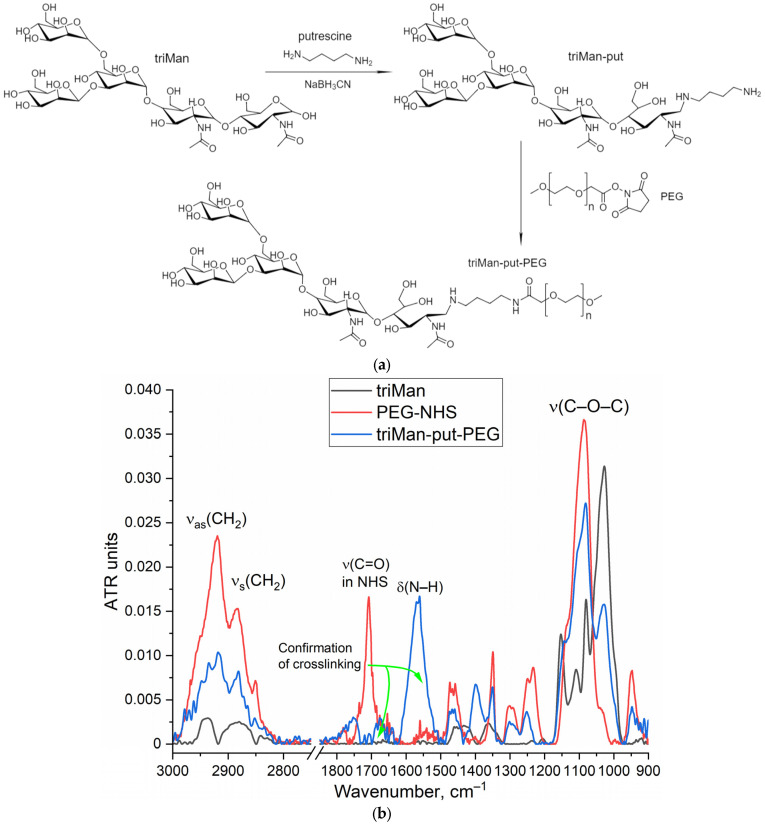
(**a**) The scheme of synthesis of the triMan–PEG IR label. (**b**) FTIR spectra of the initial substances and the target product. PBS (0.01 M, pH 7.4). T = 22 °C.

**Figure 2 biomedicines-11-02769-f002:**
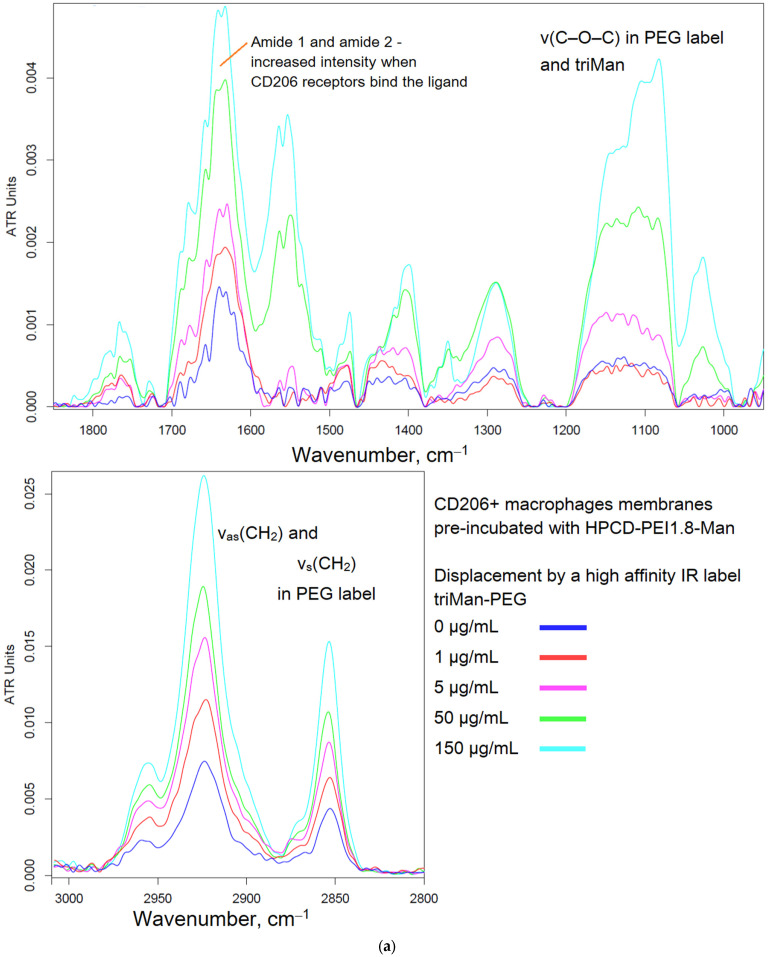
(**a**) FTIR spectra of CD206+ macrophage membranes pre-incubated with medium-affinity polymer HPCD-PEI-Man followed by displacement with high-affinity IR label triMan–PEG. (**b**) Corresponding dependences of FTIR intensities. (**c**) FTIR spectra of CD206+ macrophage membranes pre-incubated with high-affinity IR label triMan–PEG followed by displacement with high-affinity polymer HPCD-PEI-triMan. PBS (0.01 M, pH 7.4). T = 37 °C.

**Figure 3 biomedicines-11-02769-f003:**
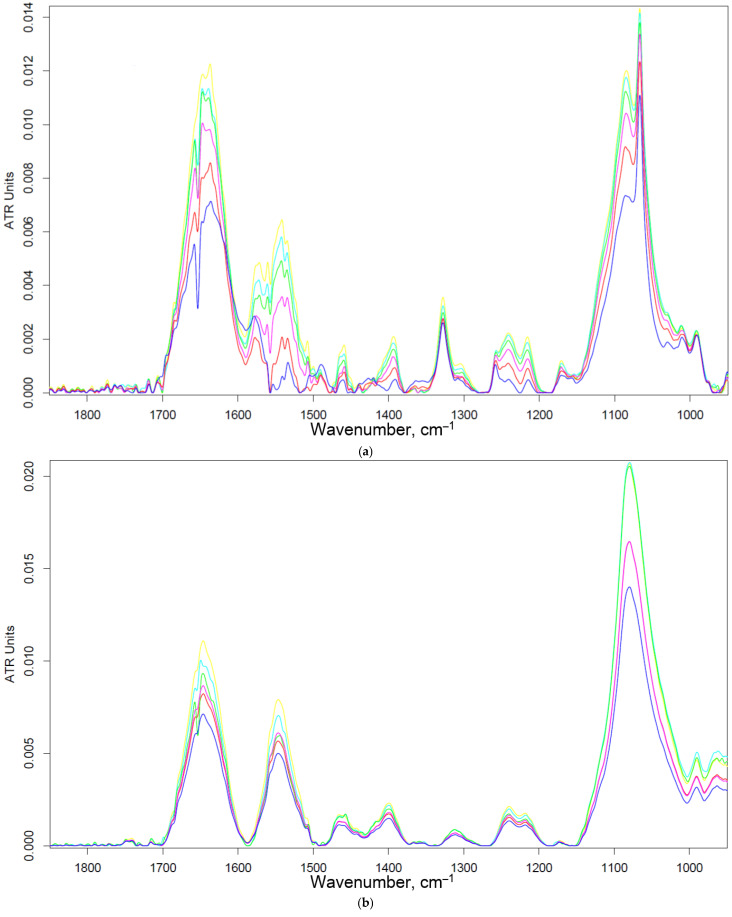
FTIR spectra of HEK293T (5 × 10^5^ cells) during online incubation with (**a**) HPCD-PEI-Man, (**b**) HPCD-PEI-Gal, (**c**) HPCD-PEI-triMan, and (**d**) Mannan-sp-HPCD: 0 min (blue), 5 min (red), 10 min (purple), 15 min (green), 20 min (cyan), 30 min (yellow) PBS (0.01 M, pH 7.4). T = 37 °C.

**Figure 4 biomedicines-11-02769-f004:**
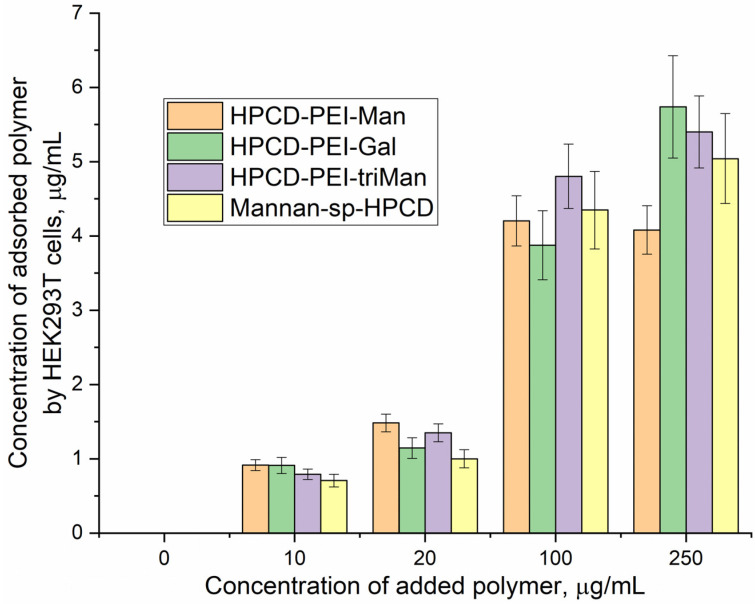
Concentrations of adsorbed polymers versus concentrations of added polymer to HEK293T (CD206−) cells. Fluorescence detection by FITC: λ_exci_ = 490 nm, λ_emi_ = 515 nm; 2 h incubation cells with FITC-labeled polymers. T(incubation) = 37 °C. T(measurement) = 22 °C.

**Figure 5 biomedicines-11-02769-f005:**
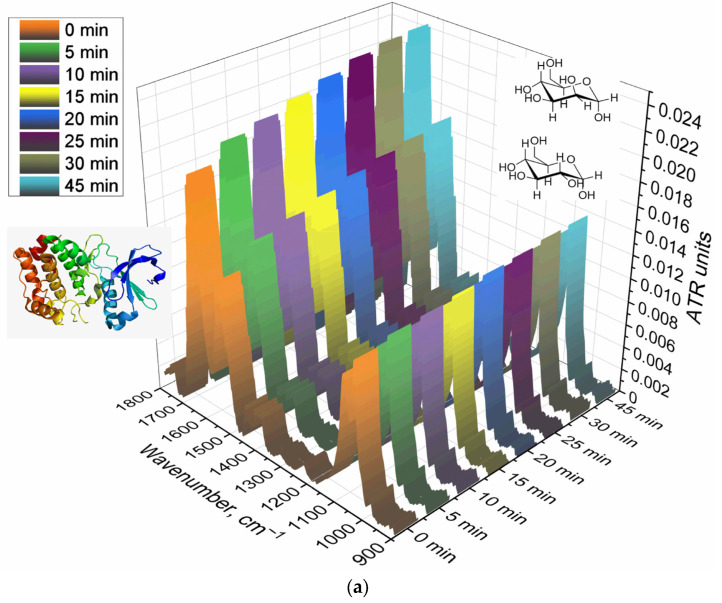
FTIR spectra of BAL suspension during online incubation with (**a**) HPCD-PEI-Man, (**b**) HPCD-PEI-Gal, (**c**) HPCD-PEI-triMan, and (**d**) Mannan-sp-HPCD: 0–45 min. PBS (0.01 M, pH 7.4). T = 37 °C. (**e**) Relative changes in Amide I peaks in the FTIR spectra of BAL after 45 min incubation with different polymer concentrations.

**Table 1 biomedicines-11-02769-t001:** Polymers for recognition by macrophages and their properties. Correlation of the binding of polymer ligands with living macrophages and macrophage-derived membranes. Differentiated CD206+ macrophage cells were studied. Macrophage-like cells were derived from THP-1 cell line.

Polymer *	Molecular Weight, kDa	Particle Size ***, nm	Affinity for Membranes of CD206+ Macrophages, % ****	Affinity for Living CD206+ Macrophages, % *****	Confocal Images of CD206+ Macrophage with FITC-Labeled Polymers
Mannan-sp-HPCD-FITC ** (1:15:10:1)	60 ± 8	240 ± 40	71 ± 3	69 ± 2, high affinity	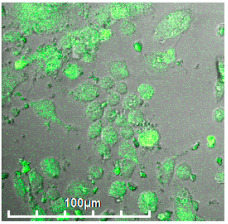
HPCD-PEI1.8-Man-FITC ** (5:1:10:0.3)	11 ± 3	180 ± 50	52 ± 2	60 ± 3, medium affinity	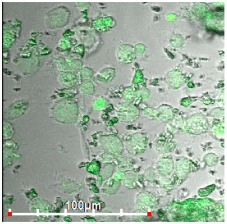
HPCD-PEI1.8-triMan-FITC ** (5:1:6:0.3)			76 ± 4	80 ± 4, high affinity	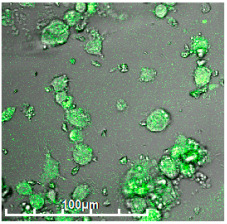
HPCD-PEI1.8-Gal-FITC ** (5:1:10:0.3)			34 ± 2	56 ± 2, low affinity	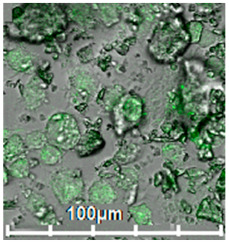

* Both FITC-labeled polymers and non-labeled polymers were used. ** HPCD—(2-hydroxypropyl)-β-cyclodextrin, sp—spermine, PEI—polyethyleneimine, FITC—fluorescein isothiocyanate. *** by Nanoparticle Tracking Analysis. **** Proportion of bound polymer. ***** Determined using flow cytometry; % of FITC-positive cells are given.

**Table 2 biomedicines-11-02769-t002:** The proportion (% from added) of the adsorbed FITC-labeled polymers by the CD206+ macrophage membranes with pre-adsorbed non-labeled polymers of different affinities for CD206 receptor.

		Membranes of CD206+ Macrophages with Pre-Adsorbed Polymer *			
Displacing Polymer *	Concentration of Displacing Polymer	Mannan-sp-HPCD	HPCD-PEI-Gal	HPCD-PEI-Man	HPCD-PEItriMan
HPCD-PEI-triMan-FITC, High affinity	5 µg/mL	14 ± 2	60 ± 4	36 ± 5	21 ± 3
	100 µg/mL	4 ± 1	29 ± 3	15 ± 2	9 ± 1
HPCD-PEI-Gal-FITC, Low affinity	5 µg/mL	8 ± 1	43 ± 5	18 ± 1	10 ± 2
	100 µg/mL	3 ± 1	17 ± 3	6 ± 1	2 ± 0.5

* The polymer with the triMan or mannan label has a high affinity for CD206 macrophages. Man—medium affinity, Gal—low affinity.

**Table 3 biomedicines-11-02769-t003:** Binding of FITC-labeled ligands to bronchoalveolar and nasopharyngeal lavages and inhibition of this interaction by mannan to elucidate the CD206-dependent mechanism.

Polymer, 0.5 mg/mL	Bronchoalveolar Lavage		Nasopharyngeal Lavage	
	Free	+Mannan	Free	+Mannan
HPCD-PEI-Man-FITC	10 ± 1	4 ± 1	12 ± 1	46 ± 3
HPCD-PEI-Gal-FITC	7 ± 1	9 ± 1	4 ± 1	54 ± 4
HPCD-PEI-triMan-FITC	27 ± 3	18 ± 2	70 ± 4	58 ± 4
Mannan-sp-HPCD-FITC	16 ± 2	9 ± 1	63 ± 3	59 ± 5

## Data Availability

The data presented in this study are available in the main text.

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
