# Peer review of "Biomimetic System Based on Reconstituted Macrophage Membranes for Analyzing and Selection of Higher-Affinity Ligands Specific to Mannose Receptor to Develop the Macrophage-Focused Medicines"

_biomedicines, 2023, doi:10.3390/biomedicines11102769_

Round 1

Reviewer 1 Report

In this manuscript, the authors report on the development of a biosensor (a mannose receptor ligand) and an a-cellular model (macrophage membrane) that should create the basis of a platform to define the status of macrophages in pathological settings, thus providing a therapeutical tool. This is a very good concept, and they propose an appropriate approach. However, although the authors are experts in designing, testing and developing biomimetics (as shown by their publication record), I am concerned with the biological system they use and present, or actually, do not really present.

Indeed:

-There is no data on the characterization of the membranes:

The authors state the system is stable, but on which experimental data?

How do I know that they are working with membranes and not with debris or cell aggregates?

-There is no characterization of the inflammatory status of the BAL or NPL that are used. The authors should provide a minimum of information, such as data on the expression levels of a restricted set of inflammatory specific genes or proteins. These data would increase the value of the results obtained with the biosensors.

-The bioassays are not well described. The way they are reported indicates that we look at internalized markers, not at markers binding at the surface, at least in living cells since all assays are done at 37°C (e.g. Fig 2 or 4). If this is the case, how can we exclude the possibility of fluid phase endocytosis in HEK cells or in living macrophages instead or parallel to receptor-specific uptake? This should be addressed in the analysis or at least in the discussion of the results.

Specific comments:

Introduction:

Lines 41-43: the sentence should be rephrased. Its meaning is not clear.

Line 65-88: I am a bit confused. I am not sure whether the authors refer only to their work, which would be the logical consequence of starting this section with  “We offer,,,, (line 65). Lines 65-68 clearly refer to their work and first application they “offer”; lines 69-76 is unclear; lines 77-81 clearly refer to the work they will report in the manuscript; lines 82-88 describe the third application the authors want to develop, and which is followed from line 89 by the description of the aim of the study. It would be good to restructure this part of the introduction.

Methods:

-THP1 cells differentiation = proof that cells are differentiated?

-HEK cells have been used in many different labs and different clones might exist in different labs: what is the proof that the HEK cells the authors use are CD206 negative?

-Obtention of membrane: the authors mention they “washed with “a” buffer” ? Which buffer? What means “dry with dry air”? The method should be described in more detail.

-Binding capacity:

Cells are detached: how? This could affect the binding capacity or number of receptors at the surface. Did the authors check the presence of MR after detachment by flow cytometry for instance, to make sure that the MR is still present ? At which temperature is done the detachment?

How do they check for “true” binding on macrophage, i.e. how do they rule out unspecific binding? Cell surface binding is usually assayed at 4°C; here the reported temperature is 37°C.

Results:

Table 1: Images: what kind of macrophages are those? THP1 ? It should be indicated in the figure legend or in the main text.  Cells look in bad shape in the last image. Did the authors check for viability of the cells (trypan blue for instance or other assay) before testing for binding?

Macrophage membrane: lines 288-293: there is no data on the membranes, except data shown for the binding in Table 1. How do the cells/membranes look like before and after drying at the air, and after storage at 4°C? How do we know these are still membranes and not debris? Where is the proof of specificity? For instance, can the authors rule out that the membranes bind to intracellular molecules/receptors that should not be available to the biosensor?

Figure 3, 4: binding on HEK cells is done at 37°C: i.e. we look at binding AND possibly internalisation via fluid phase endocytosis. Can the authors rule out this possibility?

English is fine. The authors should clarify the meaning of some sentences by rephrasing them, as mentionned in  my review.

Reviewer 2 Report

Zlotnikov et al describe interesting approaches to synthesize chemicals that bind efficiently to CD206 of macrophages and possibly to dendritic cells and some endothelial or epithelial cells. In addition, they store ligand bound membranes for months at 4 C for later analysis.

It would be interesting to visualize the stored membranes at 4 C. Are the CD206 molecules clustered or present evenly distributed on the membrane?

Is it possible to store membranes and then add the ligands?

CD206 has 13 known members. Did the authors test the binding of the novel ligand to different members of CD206?

Do these ligands cross react with mouse CD206?

Round 2

Reviewer 1 Report

I thank the authors for their reply and for the corrections they have made to their manuscript. I am satisfied with their answers.

I have a few remarks or suggestions related to typing mistakes or use of words:

Line 101: Replace “consisting” by “consists”?

Line 209: do you really mean “teased” or do you mean “tested”  ?

Line 223: “membranes”, not “membrane”

Line 316: please delete “of”

Line 392: replace “Bindind” by “binding”

Line 417: maybe replace the second “determine” by “define” ?

Lines 420-421: I would rephrase the sentence as follows: “The BAL we studied contained 3.4% of neutrophils, 87.4% of macrophages…etc.”

Line 422: I am not sure that the word “substance” is appropriate. You mean the BAL, right? Then I would propose to use “BAL”.

Line 433: replace “bonding” by “binding”

See my comments above